Spatial variations of soil seed banks in Shanghai’s urban wasteland: a gradient analysis of urbanization effects

Xia Siyi 1
Zhang Shumeng 1
Cui Yichong 1
http://orcid.org/0000-0003-0800-5968 Gao Zhiwen 1 2 3 zwgao@des.ecnu.edu.cn
Song Kun 1 2 3 ksong@des.ecnu.edu.cn
Da Liangjun 4
1 Zhejiang Tiantong National Forest Ecosystem Observation and Research Station, School of Ecological and Environmental Sciences, East China Normal University , Shanghai , China
2 Global Institute for Urban and Regional Sustainability, East China Normal University , Shanghai , China
3 Institute of Eco-Chongming , Shanghai , China
4 Institute of Science and Engineering of Ecology in Arid and Semi-arid Areas, Xi’an University of Architecture and Technology , Xi’an, Shaanxi , China
Brygadyrenko Viktor
Electronic publication date: 2024 Dec 23
Publication date: 2024
Volume: 12
Electronic Location ID: e18764
Received 2024 Sep 25; Accepted 2024 Dec 4
Copyright: © 2024 Xia et al.
Copyright year: 2024
Copyright holder: Xia et al.
License: This is an open access article distributed under the terms of the Creative Commons Attribution License, which permits unrestricted use, distribution, reproduction and adaptation in any medium and for any purpose provided that it is properly attributed. For attribution, the original author(s), title, publication source (PeerJ) and either DOI or URL of the article must be cited.
License URL: https://creativecommons.org/licenses/by/4.0/

Keywords: Seed bank, Wasteland, Urbanization, Plant diversity

Funding: The Ministry of Science and Technology of China 2015FY210200-4 Natural Science Foundation of China 31770468 This research was funded by the Ministry of Science and Technology of China (2015FY210200-4), Natural Science Foundation of China (Project No. 31770468). The funders had no role in study design, data collection and analysis, decision to publish, or preparation of the manuscript.

==============================
Urbanization greatly impacts both the diversity of soil seed banks and the spatial dynamics of species. These seed banks serve as a window into the ecological history and potential for recovery in urban wastelands, which are continually evolving due to urbanization. In this study, we selected 24 plots along urban-rural gradients in Shanghai, China. Soil samples were collected from each plot for seed bank germination experiment in both spring and autumn. We tested whether the seed density, species diversity, and composition of soil seed banks in wasteland varied along an urban-rural gradient. The results showed that seed density was higher in autumn than in spring and no significant difference was found along urban-rural gradients. A total of 74 species, belonging to 26 families and 69 genera, was recorded in soil seed banks, in which annuals were the dominant life form and autochory was the dominant dispersal model. The proportion of exotic species was nearly 40%. There is no significant difference along urban-rural gradients for functional composition, species diversity, and species composition, excepting that marginal significant for autumn Shannon-Wiener index and species composition among urban-rural gradients. The relative homogeneity in the seed bank across urban-rural gradients may primarily be due to the young age of the wastelands.

Introduction

Urbanization triggers swift alterations in the plant habitats, leading to shifts in patterns of plant diversity distribution and homogenization (McKinney, 2006; Gao et al., 2023). The intricate interplay of urban and rural gradients, shaped by human development dynamics, emerges as a pivotal factor influencing urban biodiversity and landscape configurations (Hu et al., 2020; Hou et al., 2023). Extensive research since the inception of the urban-rural gradient concept has unveiled diverse patterns and driving factors governing the distribution of species along the urbanization gradient (McDonnell & Hahs, 2008; Tian, Song & Da, 2014; Cui et al., 2019; Gao et al., 2023). This comprehensive understanding contributes significantly to unraveling the complexities of urban ecosystems, shedding light on the diverse factors influencing the distribution patterns and dynamics of plant species in response to urbanization. However, our current comprehension of how urbanization shapes urban plant diversity predominantly relies on knowledge pertaining to above-ground diversity. Knowledge regarding the characteristics of urban wastelands soil seed banks and the diversity changes induced by urbanization remains limited. Exploring these aspects is crucial for a more holistic understanding of urban ecosystems and can offer valuable insights into the intricate dynamics of plant life within urbanized environments.

A soil seed bank represents the cumulative presence of all viable seeds on the surface and in the soil layers beneath a specific area (Yang et al., 2021). The diversity of soil seed banks is intricately linked to seed sources from current and past vegetation, providing a partial record of the area’s ecological history (Engel & Martins, 2007; Wu et al., 2024). In various vegetation systems, seeds remain dormant in the soil, with their individual numbers may surpassing those of aboveground vegetation (Lamont & Pausas, 2023). They contribute to population reestablishment through the ‘rescue effect’ (Veken et al., 2007) and facilitate species coexistence via the ‘storage effect’ (Chesson, 2000). In this way seed banks can buffer plant populations against unfavorable environmental conditions and stochastic disturbances (Chesson, 2000) which may contribute to time-lags in the response of aboveground vegetation to rapid and increasing rates of global environmental change (Plue et al., 2020). Therefore, these seed banks serve as a critical source for the growth and reproduction of plant communities, shaping the establishment and succession of vegetation, and potential contributors to plant species diversity (Vandvik et al., 2016). Thus, it is reasonable to understand how seed banks vary in the context of urbanization, which will help us to take insight of the effects of urbanization on plant diversity.

Land use in urban areas often experiences significant transformations, with urban wasteland enduring relatively infrequent human disturbances. These wastelands host communities aligned with the local climate and disturbance context (Cui et al., 2019). They represent an excellent location for studying the relationship between urbanization and plant diversity (Anderson & Minor, 2019). Urban flora’s central reservoir comprises the majority of spontaneous plants found in these wastelands (Maurer, Peschel & Schmitz, 2000; El-Ghani et al., 2011; Robinson & Lundholm, 2012). Moreover, urban wastelands can act as stepping stones for species dispersal and interaction (Wittig, 2012). They not only function as hotspots for urban biodiversity (Angold et al., 2006; Bonthoux et al., 2014) but also provide temporary refuge for endangered species (Albrecht et al., 2011; Meffert, Marzluff & Dziock, 2012). The crucial role of urban wastelands in conserving urban biodiversity is increasingly acknowledged by researchers (Eckert et al., 2017; Rega-Brodsky, Nilon & Warren, 2018). While extensive research has focused on the aboveground species diversity, functional diversity, and phylogenetic diversity in these abandoned spaces (Bonthoux et al., 2014), the composition characteristics of the soil seed bank in urban wastelands and its distribution pattern along the urbanization gradient remain unclear.

In recent years, China has become one of the fastest growing and largest urbanized countries in the world (Gu, 2019). Shanghai, being one of the swiftest urbanizing cities in both China and globally, has dynamically produced a substantial number of urban abandoned lands throughout its urbanization progress. Those abandoned land posseted abundance spontaneous plants (Tian, Song & Da, 2014). Specifically, we hypothesis that rapid urbanization in Shanghai has influenced the seed density, species diversity, and composition of soil seed banks in urban abandoned lands, with variations along urban-rural gradients. We expect seed density and diversity are higher in rural areas and lower in highly urbanized areas due to changes in land use and environmental conditions. Our investigation focused on examining soil seed banks in the wastelands of Shanghai during in spring and autumn. Soil samples were collected from 24 plots along urban, suburban and rural gradients to explore the impact of urbanization on the diversity of weed seed banks. The objectives of this study were to: (1) Investigate the composition characteristics of soil seed banks in wasteland under the background of urbanization; (2) Test whether seed density, species diversity and composition of soil seed bank in wasteland varied along urban-rural gradients.

Materials and Methods

Portions of this text were previously published as part of a preprint (https://doi.org/10.21203/rs.3.rs-4485892/v1).

Study site

This study was conducted in Shanghai (30°40′–31°53′N, 120°52′–122°12′E), which is one of most developed cities in China. Its annual average temperature is 18.0 °C and the annual total precipitation is 1,044.1 mm (Shanghai Municipal Statistics Bureau (SMSB), 2023). More than 81% of the annual rainfall is concentrated between April and October. The city is approximately 6,340.5 km2. Since 1970s, Shanghai experienced rapid urbanization. Except for less 1% remnant vegetation, the vegetation type in Shanghai is dominated by weeds community and cultivation vegetation (Han, 2020).

Sampling methods

The city of Shanghai expands to become more rural from the central urban area in the form of concentric circles. Therefore, the city can be divided into three differing levels of urbanization areas (i.e., urban, suburban and rural) according to its two major urban ring roads (please see Fig.1), as we did in previous study (Tian, Song & Da, 2014). Along the urban-rural gradient, survey sampling sites are radially established. Our study selected 24 wasteland sites for soil seed bank investigations, including seven sites in urban area, 13 sites in suburban areas, four sites in rural areas (please see Table 1). The sites in our study are publicly-owned land, which all can get access to enter. All wasteland sites had between 1 and 15 years of abandonment, and there is no significant difference in abandoned time among sites of the urban-rural gradient (p > 0.05).

Figure 1 Location of the study area and the distribution of sampling sites.

Table 1 The location of wasteland sites and the number of soil samples.

Site	Gradient	Longitude (°E)	Latitude (°N)	Number of soil samples	
Autumn	Spring	
1	Suburban	121.4549	31.04613	0	15	
2	Suburban	121.5169	31.06907	0	15	
3	Urban	121.4491	31.23803	15	15	
4	Urban	121.4322	31.19365	15	0	
5	Urban	121.4096	31.19393	15	15	
6	Urban	121.5175	31.32790	15	15	
7	Urban	121.5290	31.17400	15	15	
8	Suburban	121.5087	30.96397	15	15	
9	Rural	121.5207	30.91534	15	15	
10	Suburban	121.6068	31.10245	15	15	
11	Suburban	121.7329	31.04350	15	15	
12	Rural	121.8068	30.97146	15	15	
13	Rural	121.9325	30.88532	15	15	
14	Suburban	121.4521	31.11623	15	15	
15	Suburban	121.3286	31.09917	15	0	
16	Suburban	121.3227	31.03025	15	15	
17	Suburban	121.2349	30.98634	15	15	
18	Suburban	121.3097	31.30171	15	15	
19	Suburban	121.1976	31.33710	15	0	
20	Suburban	121.3140	31.21783	15	15	
21	Rural	121.1525	31.15058	15	15	
22	Urban	121.5016	31.21558	15	0	
23	Urban	121.6291	31.23161	15	15	
24	Suburban	121.4178	31.36050	15	15	

To ensure thorough soil sample collection, a 10 × 10 m plot was selected in the center of each wasteland site and each 100 m2 plot was further divided into 100 quadrats by 1 m × 1 m. We randomly selected five quadrats for soil sample collection each time, from which three soil samples were taken and thoroughly mixed for subsequent germination experiments. The sampling was conducted in both autumn and spring to capture seasonal variations in seed bank composition, density, and diversity. Autumn is typically a period when many plant species release their seeds, providing a peak in seed bank replenishment. Spring, on the other hand, is when germination typically begins, allowing us to assess the presence of viable seeds and the potential for regeneration. By sampling across these two seasons, we aimed to capture a comprehensive picture of the seed bank dynamics throughout the year, accounting for both seed deposition and germination patterns. Therefore, soil samples were collected from the same quadrats in autumn (October to November) 2015 and spring (May to June) 2016. A cylindrical soil sampler with a diameter of 5 cm is used to drill a 15 cm deep soil sample (Shang et al., 2012). In total 22 sampling plots were collected during autumn (two sites have an excessive amount of gravel and debris), and 20 plots (four sites were already been used for other purpose) were collected during spring survey.

The soil’s physical and chemical properties can directly impact seed banks by influencing seed germination and aging through soil water-holding capacity, or indirectly by affecting seed viability through soil pathogen activity (Yang et al., 2021). For investigating soil characteristics, a cylindrical soil sampler with a diameter of 5 cm is used to drill a 15 cm deep soil sample in September 2015. Soil pH was measured at a soil: water ratio of 1:2.5 (weight/weight). Air-dry soil and deionized water were shaken together then pH determined with a pH electrode. Soil hydrolyzable nitrogen (SHN) were determined by the detection of ammonium. Soil samples were extracted with concentrated sulfuric acid and hydrogen peroxide. Soil available phosphorus (SAP) concentration was determined after the detection of phosphate by extraction with sodium bicarbonate (Zheng et al., 2020). The soil characteristics are summarized in Table 2.

Table 2 Soil characteristics on urban-rural gradient.

Soil characteristics	Urban	Suburban	Rural	
pH	8.2 ± 0.4a	8.2 ± 0.2a	8.1 ± 0.1a	
SHN (mg/kg)	112.6 ± 19.5a	144.9 ± 129.3a	104.4 ± 32.0a	
SAP (mg/kg)	9.8 ± 1.8a	8.9 ± 1.4a	10.2 ± 2.6a	
Note:

The letter ‘a’ indicate non-significant differences (p > 0.05) of soil characteristic on urban-rural gradients, based on the Wilcoxon Rank-Sum Test. SHN, soil hydrolysable nitrogen; SAP, soil available phosphorus.

Soil seed banks quantification

The seed bank was evaluated using the seedling emergence method as described by Simpson, Leck & Parker (1989), with experimental conducted in a greenhouse. We effectively terminated the seeds’ dormancy by employing a combination of drying processes and low-temperature storage techniques (Mo, 2013). This method allows us to capture the germination of seeds present in the soil, providing a direct measure of seed bank content without disturbing the soil structure (Hartzler & Smidt, 1993; Mo, 2013). Seedling emergence is particularly useful for identifying viable seeds, as it reflects the presence of seeds capable of germinating and establishing new plants. Additionally, it allows us to monitor the temporal dynamics of seed bank composition over time, which is important for understanding how urbanization and environmental factors influence seed bank characteristics. Following the above-mentioned treatment, the collected soil samples, were carefully placed in a plastic germination container (37 cm × 27 cm × 5.5 cm, Fig. 2). This container was pre-lined with a 5 cm layer of sterile coconut shell subsoil to ensure aseptic conditions. The Autumn germination experiment begun in February and concluded in June 2016, while the spring germination experiment spanned from August to December 2016. Each germination time was approximately four months long to ensure there was little further germination, with regularly watering to maintain optimal soil moisture. Throughout, seedlings were diligently identified, documented, and removed on a weekly schedule, following the protocol established by Albrecht et al. (2011).

Figure 2 Seed germination experiment design.

Data analysis

Species identification and classification

The scientific names of plants were referenced from Flora of China (Editorial Committee of Flora of China, Chinese Academy of Sciences, 1991). Species were categorized as native or exotic according to the “Plant Directory of Five Provinces and One City in East China” (Zhang & Lai, 1993) and the “Wild Higher Plants Atlas of Shanghai Urban Area” (Qin, Pei & Wang, 2016). Germinating species were classified by life forms as annual or perennial, with annuals further categorizing into winter and summer annual (Zhang, 2017). Growth types were divided into eight groups (Table 3), with additional classification by height (Numata & Yoshizawa, 1988; Kitazawa & Ohsawa, 2002) (see examples in Fig. S1). Dispersal modes were divided into anemochory, autochory, hydrochory and zoochory, referencing the Pesticide Control Institute of the Ministry of Agriculture of the People’s Republic of China, Japan Plant Regulators Research Association (2000), Guo & Zheng (2017), and Lososová et al. (2023).

Table 3 Classification of species growth forms.

Growth types	Trait	Representative species	
Small growth form			
Procumbent	Extends creeping stems and root from various places	Euphorbia humifusa	
Rosette	Has radial basal leaves and leafless flowering stems	Taraxacum mongolicum	
Branched	The lower part of the stem has many branches and no obvious spindle	Stellaria media	
Tussock	Forms the root first and then grows thickly	Poa annua	
Climbing or liane	Has a stem that curls or clings to something	Humulus scandens	
Large growth form			
Partial rosette	Shows a rosette in the early stage and becomes erect after the rosette withers	Sonchus oleraceus	
Pseudo-rosette	Shows a residual rosette and has leaves on an erect stem	Youngia japonica	
Erect	The spindle of the aerial part of the plant has an obvious erect posture	Chenopodium album	

Seed density and species diversity

Seed density expressed as the number of germinated seeds per unit area (1 m2) of soil.

Two indices were used to measure the diversity in each plot: species richness (S) and the Shannon-Wiener index of diversity (H) (Magurran, 1988),

H=−∑i=1s⁡Piln⁡Pi

where Pi is the individual proportion of a species, and S is the total number of species.

To calculate the diversity index, the diversity function in the Vegan package (version 2.6) (Oksanen et al., 2022) was used. The significant differences of the seed density and species diversity indices were tested in different regions while using a Wilcoxon Rank-Sum Test (Dubitzky et al., 2013). The calculations were run in R 4.2.1 (R Core Team, 2022). Chi-square test was used to examine the significant differences in the proportion of different types on the urban-rural gradient. Using SPSS software (version 23.0) for Chi-square testing (IBM Corp., Armonk, NY, USA).

Similarity among the areas of the gradient

To test if species composition differed among the urban, suburban and rural gradients, we performed a Permutational Multivariate Analysis of Variance Using Distance Matrices (PERMANOVA), using the abundance-based euclidean index as our measure of dissimilarity (Anderson, 2001). The PERMANOVA partitions compositional distance matrices among sources of variation and fits linear models to distance matrices while using a permutation test with pseudo-F ratios to calculate clear effects. Then, non-metric multidimensional scaling (NMDS) was used to visualize the overall differences in species composition along the urban-rural gradient (He et al., 2020). The PERMANOVA and NMDS were run using the R package vegan (Oksanen et al., 2022).

Results

Seed density

In autumn 2015, the density of germinated seeds was 2,675 seeds/m2. In spring 2016, the density of germinated seeds was 837 seeds/m2. The soil seed density in autumn is higher than that in spring, and there is no significant difference in seed density among sites of the urban-rural gradient in different seasons (p > 0.05) (Table 4).

Table 4 Seed density on urban-rural gradient.

Sample time	Urban (seeds/m2)	Suburban (seeds/m2)	Rural (seeds/m2)	
Autumn 2015	3,390 ± 4,509a	2,306 ± 1,642a	2,487 ± 1,487a	
Spring 2016	636 ± 366a	923 ± 557a	855 ± 617a	
Note:

The letters ‘a’ indicate non-significant differences (p > 0.05) of soil characteristic on urban-rural gradient, based on the Wilcoxon Rank-Sum Test.

Assemblage structure, growth type and dispersal modes

In total, we recorded 74 germinated plant species, belonging to 26 families and 67 genera (Fig. 3), The germinated species list in Table S1. Autumn and spring seasons respectively yielded 59 and 55 species, with 40 species observed in both. The family Poaceae (13 genera and 13 species) was the most frequently, followed by Asteraceae with 11 genera and 11 species. Of the total plants, 46 (62.3%) were native species and 28 (37.8%) were exotic species. Annual plants (75.7%) were the dominant life-form, including 23 (31.1%) winter annual species and 33 (44.6%) summer annual species (Fig. 4A). Small growth type species made up 58.1 % of the total, with tussock being the most common (13 species), followed by branched (10 species), climbing or liana (nine species), procumbent (seven species), and rosette (four species, Fig. 4B). Large growth type species constituted 41.9 % (31 species), with erect species being the most prevalent (15 species), followed by partial rosette (nine species) and pseudo-rosette (seven species). Of all germinated plant species, the major seed dispersal type is Autochory (35 species, 42.3%, Fig.4C).

Figure 3 The heatmap of weed species in soil seed bank of wastelands.

Light gray represents that the species did not occur in this sampling site. Below the image is the survey season and the name of the site; on the left side of the image are the species Latin names.

Figure 4 Species composition characteristics of weed soil seed banks along the urban-rural gradient.

(A) Life form composition, (B) growth type composition and (C) dispersal modes composition.

Across both seasons, the urban-rural gradient revealed no significant differences in the distribution of life forms, the proportions of native vs exotic species, the proportions of each growth type, or the proportions of each dispersal mode (p > 0.05).

Species composition and richness

Species composition differed among the three urban-rural gradients (PERMANOVA: F (2,19) = 1.436, p = 0.045, R2 = 0.131) in autumn, but did not significant differed in spring (PERMANOVA: F (2,17) = 0.876, p = 0.693, R2 = 0.093). From two-dimensional NMDS ordination, urban-rural gradients differed in terms of species assemblages with small overlap in autumn, while there is large overlap among three urban-rural gradients (Fig. 5).

Figure 5 Two-dimensional nonmetric multidimensional scaling (NMDS) ordination of species composition of soil seed bank along an urban-rural gradient in Shanghai.

Ordination of seed bank was based on species abundance data. The size of the circle indicates richness.

In autumn, the mean value of richness and Shannon-Wiener index of each plot is 11 and 1.7 in autumn respectively, while they are 11 and 2.0 in spring respectively. Wilcoxon rank-sum test showed that urban exhibited the lowest Shannon-Wiener index in autumn, which was significantly lower than the values found for rural (p = 0.047). However, there is no significant difference along urban-rural gradients in spring (p > 0.05) (Fig. 6).

Figure 6 Boxplots of species richness (A) and Shannon-Wiener index (B) for soil seed bank of wastelands along the urban-rural gradient.

Capital letters and lowercase letters indicate significant (p < 0.05) differences in autumn and spring, respectively, based on the Wilcoxon rank-sum test. Box plots indicate median (middle line), 25th, 75th percentile (box), and 5th and 95th percentile (whiskers) as well as outliers (single points).

Discussion

Soil seed banks are vital for the long-term survival of the diversity and dynamics of plant communities (Bakker, 2001). The presence of wastelands in cities contribute significantly to the plant diversity in urban areas (Godefroid, Monbaliu & Koedam, 2007). However, the dynamic in space of soil seed banks in wasteland during the process of urbanization is poorly documented.

In our study, the seed density in urban wasteland of Shanghai is much lower than urban wasteland in Munich, German (Albrecht et al., 2011) and Tianjin, China (Mo, 2013). This discrepancy may be due to the longer time span since abandonment compare to our study. We speculate that as the duration of abandonment increases, more species may be able to produce stable seed banks, which could consequently promote an increase in the number of species within the soil seed bank. The urban wasteland Tianjin has been abandoned for 40 years. But in this study, the abandonment time of the study site was no more than 15 years. An additional factor contributing to these differences could be the impact of temperature and precipitation. Munich experiences an average annual temperature that is roughly 10 °C cooler than Shanghai’s, and it receives about 100 mm less annual rainfall. These climatic variables can significantly influence seed dormancy-breaking and germination, as well as the metabolic activity of seeds and soil fungal activity, which in turn can affect the density and richness of the seed bank (Baskin & Baskin, 2014; Yang et al., 2021). Additionally, when comparing with formal urban greening, the urban wastelands, an informal urban greening, own higher seed density. For example, compare to our study, the soil seed density of urban forest parks in Chongqing (Xu, 2016), Beijing (Xue, Zhao & Sun, 2023), Shenyang and Fushun (Zhang & Gao, 2009) showed lower seed density. The main reason could be that in formal urban green spaces, there’s a significant amount of human management, making it difficult for weed populations to form stable communities.

The seed density in urban wasteland exhibits distinct seasonal variation, with seed density in spring being notably lower than that in autumn. Our autumn sampling was in November, when the seeds of most summer annual and perennial plants got mature and many of them forming a transient seed bank with a large proportion of germinable seeds (Thompson & Grime, 1979). The sampling in spring was in April, when the seeds in the transient seed bank have already germinated, resulting in a lower seed density.

Our study found that the seeds bank of wasteland in Shanghai are dominate by annuals plants and small growth type species. Many studies about soil seed bank have found similar results (Fischer et al., 2013; Fu, 2016; Xu, 2016; Li, Dong & Guan, 2018; Xue, Zhao & Sun, 2023). Annuals plants and small growth type species are generally highly competitive and plasticity, and can survive in the soil for a long time. The dominance of annuals in the seed bank of Shanghai’s wastelands, as observed in our study, suggests a rapid turnover of plant species that may be indicative of the dynamic and transitional nature of these urban ecosystems (Poppenwimer, Mayrose & DeMalach, 2023). Rapid urbanization process leading a higher intensity and frequency human interference and resulting in plant communities difficult to succeed to the perennial stage (Li, 2009). The prevalence of small growth type species in the seed bank could be attributed to their adaptive advantages in wastelands, such as requiring less space and resources, allowing them to thrive in the potentially nutrient-poor and disturbed soil conditions often found in these areas. In addition, plants of small growth types that crawl and grow have strong adaptability to trampling, maintaining their normal reproductive and transmission abilities, and surviving in urban areas (Tian et al., 2008). As observed in our study, there are no significant differences across the urban-rural gradient in the distribution of life forms and growth types. This indicates a certain level of ecological homogeneity in the studied wastelands. These wastelands may not have experienced the same environmental pressures or land-use changes that typically drive shifts in species distributions and they have similar microclimatic conditions or soil properties. These environmental factors are beneficial for a particular set of growth forms, regardless of the urban or rural status of the site.

It is generally believed that in the early stages of succession or frequently disturbed habitats, the dominant species of vegetation are mostly anemochory, whose seeds are mostly small and dormant, making them easy to bury in soil seed banks (Liu et al., 2022). Albrecht et al. (2011) found that the number of anemochory and zoochory dominated urban wasteland soil seed banks. Our study showed that the major seed dispersal type of soil seed bank is autochory and anemochory. This indicates that the wasteland in this study is in the early stage of succession and is subject to increased interference. However, due to the short abandonment period, the species with zoochory dispersal mechanisms have not yet formed a stable seed bank but favor autochorous plants. Additionally, seeds from autochorous plants are typically larger and more viable in stressful environments, which could also be one of the reasons they dominate. The larger seed size may provide more resources for the developing embryo, increasing the seed’s chances of successful germination and establishment, especially in the challenging conditions often found in urban wastelands (Westoby, Leishman & Lord, 1996).

Our research revealed that native species maintain a dominant position, and there is no significant difference in the proportions of native vs exotic species along the urban-rural gradient. Similar results have been found in the research of Albrecht et al. (2011) and Zhao et al. (2023), where the proportion of native species in soil seed banks is higher than that of exotic species. However, Albrecht showed that exotic plant species (after 1500 AD) accounted for 22.6% of the seeds in soil seed bank in Munich, which below this research in Shanghai and Zhao’s research in Beijing (Albrecht et al., 2011; Zhao et al., 2023). This may be due to latitude and the level of urbanization. Previous studies showed that cities at higher latitudes tend to harbor fewer exotic plant species and cities in subtropical ecoregions harbored more exotic species than cities in other ecoregions (Pyšek, 1998; Gao et al., 2021, 2023). In addition, cities with a higher urbanization level have the highest richness of exotic plant species (Aronson et al., 2014; Rat et al., 2017; Jehlík, Dostálek & Frantík, 2019). Within urban confines, the enhanced connectivity of a city can facilitate the spread of exotic species (Potgieter et al., 2024). Aronson et al. (2017) found that river increase the spread of exotic plants within a city. Therefore, within-city connectivity potentially obscuring the distinction of the urban-rural gradient.

Different from our previous study’s finding that plant species diversity followed a unimodal curve along the urban-rural gradient in Shanghai (Tian, Song & Da, 2014), we did not find clear trend of species diversity of seed bank of wasteland soil along urban-rural gradient. This very low similarity between soil seed banks and the standing vegetation has been widely recognized (Moore, 1980; Vandvik et al., 2016; Yang et al., 2021). In addition, the life span of wastelands is highly variable when facing numerous developmental pressures (Harrison & Davies, 2002). In the short to mid-term, wastelands are partly protected from human-induced disturbance by their negative image, which may favor biodiversity (Godefroid, Monbaliu & Koedam, 2007). This suggests that the response of seed banks to urbanization lag with changes of aboveground plants as it can offset fluctuations in seed yield and quality when the population is disturbed (Pakeman & Small, 2005; Doroski et al., 2020). Within the current urbanization context of Shanghai, the effects of environmental shifts and anthropogenic disruptions on plant communities can be somewhat mitigated by the presence of soil seed banks. In addition, due to the random management of wasteland (Sukopp, 2002), the ability of urban wasteland soil seed banks to resist interference may change with the time of abandonment.

From a theoretical standpoint, grasping the resilience of soil seed banks to disturbances is essential for modeling their contributions to future plant diversity and for gaining insights into urban ecological dynamics. Empirically, soil seed banks play a significant role in ecological restoration efforts (Morimoto et al., 2017). A key objective of urban vegetation restoration is to steer the composition of urban plant species towards “native-dominated” plant communities (Oldfield et al., 2013; Pregitzer et al., 2019). Consequently, urban ecological restoration initiatives can benefit from focusing on the disturbance resistance of soil seed banks in abandoned lands, which could facilitate the localization and ecological restoration of urban vegetation. Harnessing the potential of soil seed banks in these areas can bolster the resilience and recovery capacity of urban ecosystems, offering tangible benefits for urban ecological restoration and the conservation of biodiversity.

Conclusions

This study focuses on difference of weed seed bank in wasteland of Shanghai along urban-rural gradient. We did not find any significant difference along urban-rural gradients for functional composition (i.e., the proportion of annual plant species, small growth type species, exotic species, and dispersal modes), species diversity and species composition, excepting that marginal significant for autumn Shannon-Wiener index and species composition among urban-rural gradients. It suggests that the current extent of urbanization in Shanghai has not exerted a notable impact on seed density or the composition of the soil seed bank. It is important to consider that the relative homogeneity in the seed bank characteristics we observed could be a transient state influenced by recent environmental changes or disturbances. Therefore, long-term monitoring and comparative studies with other urban ecosystems would be necessary to determine whether these patterns persist over time or are subject to change.

Supplemental Information

Supplemental Information 1 Code.

Supplemental Information 2 Classification of species growth forms.

(1) Procumbent; (2) Rosette; (3) Branched; (4) Tussock; (5) Climbing or liane; (6) Pseudo-rosette; (7) Partial rosette; (8) Erect. 1-5 belong to small growth form and 6–8 belong to large growth form. Photos by Luo Xinyi and Gao Zhiwen.

Supplemental Information 3 The germinated species list of wastelands in Shanghai.

1Growth type: p, Procumbent; r, Rosette; b, Branched; t, Tussock; l, Climbing or liane; ps, Pseudo-rosette; pr, Partial rosette; e, Erect. 2Season: spring/autumn, the specie only germinated in season of spring or autumn; both, the specie germinated in both seasons.

Supplemental Information 4 Dataset 1.

Supplemental Information 5 Dataset 2.

Supplemental Information 6 Dataset 3.

Supplemental Information 7 Dataset 4.

Supplemental Information 8 Dataset 5.

Supplemental Information 9 Dataset 6.

Supplemental Information 10 Dataset 7.

Supplemental Information 11 Dataset 8.

We thank Xiushan Leng for helping with the seed germination experiment design diagram. We extend our gratitude to Zimeng Chen and Zichao Zhou for their assistance in creating the heatmap of weed species in the soil seed bank of wastelands. We also greatly appreciate their valuable suggestions, which have significantly enhanced the quality of our manuscript.

Additional Information and Declarations

Competing Interests

Author Contributions

Data Availability

The authors declare that they have no competing interests.

Siyi Xia conceived and designed the experiments, performed the experiments, analyzed the data, prepared figures and/or tables, authored or reviewed drafts of the article, and approved the final draft.

Shumeng Zhang conceived and designed the experiments, performed the experiments, analyzed the data, authored or reviewed drafts of the article, and approved the final draft.

Yichong Cui conceived and designed the experiments, performed the experiments, analyzed the data, authored or reviewed drafts of the article, and approved the final draft.

Zhiwen Gao analyzed the data, prepared figures and/or tables, authored or reviewed drafts of the article, and approved the final draft.

Kun Song conceived and designed the experiments, analyzed the data, prepared figures and/or tables, authored or reviewed drafts of the article, and approved the final draft.

Liangjun Da conceived and designed the experiments, authored or reviewed drafts of the article, and approved the final draft.

The following information was supplied regarding data availability:

The code and raw data are available in the Supplemental Files.

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
