# Peer review of "Spatial variations of soil seed banks in Shanghai’s urban wasteland: a gradient analysis of urbanization effects"

_PeerJ, doi:10.7717/peerj.18764_

## Round 0.1 · original submission · Major Revisions

Dear authors, This article touches upon a practically important issue for preserving urban biodiversity. The data seem to have been collected using the correct methodology, but I am not sure that you have analyzed all the data you have obtained. The diagrams and tables are very simple, although you should have received large arrays of data for many common plant species. Where are they? Readers want to see differences in the reactions of different plant species (at least several species, the most common species). It is impossible to talk about plants in general, since many of them have completely different life strategies. The article as a whole leaves a feeling of incompleteness, understatement. I ask you to add tables or, better yet, diagrams in the results section, designed according to the sample in Figure 5 (box analysis). I also ask you to add two tables to the Material and Methods. The first table should illustrate in as much detail as possible the location and number of biological samples taken into account in the sample plots. The second table should characterize the species composition of the plants you found, and also show their distribution in all the sample plots. I hope that such additions and careful corrections to the manuscript in accordance with the reviewers' comments will allow the new version of your manuscript to be accepted for publication.

Reviewer 1 ·

Basic reporting

The authors presented the results of research on spatial variations of soil seed banks in Shanghais urban wasteland. Most people consider urban wastelands to be useless and invisible, hence they are called urban blind spots. However, during the Covid-19 pandemic, urban wastelands were re-discovered and appreciated by city dwellers, as the Polish government's restrictions limited access to urban parks.Urban wastelands have great potential for promoting biodiversity. Thus, research on wastelands, including urban wastelands, is extremely valuable. The authors examined whether seed density, species diversity and composition of soil seed banks in wastelands changed along an urban-rural gradient.
The title of the paper is in accordance with the topic of the article. The abstract is ok. Keywords also.
I have no objections to the structure of the article and also to the literature list.
All figures and tables are informative.
The work is interesting and a good read. The results were comprehensively discussed.

Experimental design

The topic of the article fits in with the scope of the journal.
The research was properly planned and conducted.
The research methods are described in detail.

Validity of the findings

The authors have included extensive supplementary material.

·

Basic reporting

The manuscript describes the abundance and diversity of species in the soil seed bank of wastelands along an urbanization gradient (urban, suburban, rural) in Shanghai, China, in two seasons of the year (autumn and spring). The objective is to understand how seeds banks vary in the context of urbanization. English language requires revision throughout.

Although the work implicitly faces an interesting research problem (i.e. how does urbanization affect the seed bank in a same type of habitat (wastelands) along a disturbance gradient?), it lacks an explicit and relevant research question. The approach is then basically descriptive: a profound-theoretical framework is not exposed nor are expectations about the effect of urbanization, according to biologically plausible hypotheses posited to answer a research question.

Descriptive studies might contribute to scientific knowledge, and it is not my intention to enter into that controversy. However, a problem that descriptive studies cannot avoid is that, by not presenting specific expectations (hypotheses, predictions), the decisions made regarding sampling design cannot be fully understood and assessed a priori (e.g. Why was the sampling carried out in autumn and spring? Why was the method of seedling emergence used? Could this technique has biased the results in a way that the inferences made have been affected? Why the soil analyses were carried out? ETC.). Some of these decisions, notwithstanding, may be considered justified by assertions in the Discussion section…

Literature reviewed seem sufficient since studies of seed bank dynamics in cities are scarce.

The sampling design and techniques employed to gather seed bank information seem standard and correct, although more details on some techniques are needed (e.g. on the treatments used to break dormancy; could there have been a certain bias in favor of the emergence of some species?, e.g. annual plants). Further, I could not understand some assertions in the Method section (e.g. lines 118-120). More detailed information is needed to make the study replicable.

Data analysis appears simple and correct. I am not completely familiar with PERMANOVA to assess differences and similitudes of species composition, but it could be a fine way to doing the task.

Tables and figures summarize the main results quite clearly. However species-specific seed abundances (i.e. raw data) should be included for each occasion and area, as a table or as supplementary information.
Since the study is descriptive, the authors justify it because the dynamics of seed banks in cities is little known. Thus, the Discussion usually takes the form of a [new] exposition of the results, this time with comments that deepen their “likely” meaning. Although several results are interesting (e.g. the functional homogeneity of soil seed banks in urban, suburban and rural areas; prevalence of annual plants, etc.), their importance would have been highlighted if the authors had preferred to use them to answer challenging questions, both theoretically and empirically significant.
Some results that contrast with others from similar studies in other cities are justified by the short abandonment period of Shanghai wastelands. However, the abandonment period of wastelands in, for example, Munich does not seem to justify such a conclusion. The causes of that and other patterns could be better guessed by turning to other possible explanations.

Experimental design
* * *
Validity of the findings
* * *
Additional comments
* * *
Reviewer 3 ·

Basic reporting

Dear authors,
Overall, your manuscript is well organized and follows the Instructions for authors recommendations.
The English language is fine, no issues detected. Figures and tables are relevant and necessary, raw data are also provided as supplementary files. Results and Discussion are thorough. References are adequate, but you should made a double check of the reference list as some of them need technical corrections (they do not meet the instructions guidelines).

Experimental design

The aim of the study could be much more strenghtehed. A research hypothesis will be useful also, if you define such.
Please, cite dome references for the methodology used for the soil analyses - pH, SNH, SAP

Validity of the findings

The results obtained are interesting and meaningful. The statistical evaluation is well done.
The manuscript lacks a robust conclusion that have to highlight the new findings and their scientific significance.

Additional comments

1) Line 101, 273: Shanghai with capital letter
2)Line 112-113: Please, unify the symbol used - 10×10m and 1m * 1m
3) Line 117 and 121: Please, unify the description of soil sampler - "soil sampler with a diameter of 5 cm is used to drill a 15 cm deep soil sample" or "soil sampler with a diameter of 50 mm is used
122 to drill a 20 cm deep soil sample". Why the deep is different - 15 or 20 cm?
4) Line 130-131: Whish one is correct "as described by Simpson et al." or "by (Simpson, Leck & Parker, 1989)"?
5) You have to check the citations into the text. In case of more than one reference, they should be ordered chronologically. So, you have to correct where needed, i.e. Line 45, Line 74, etc.
6) it willbe usefull to provide a legend below Table 1, explaining the abbreviaytions used (SNH, SAP)
7) Table 2: Please, check the name of Chenopodium album (it seems to be 1 instead of l)
8) Table 3: Please, add the measuring units. Isn't there some significant differences found? Why all values are marked with the same letter "a"?
9) Fig. 3: there should be a title before a, b and c. Probably " Plant performance" or something else.
10) Fig. 5 b: Shannon with capital letter

---

## Round 0.2 · accepted · Accept

Dear authors, I am pleased to inform you that your article has been accepted for publication.

Reviewer 1 ·

Basic reporting

no comment

Experimental design

no comment

Validity of the findings

no comment

Additional comments

I see that the authors have revised the manuscript as recommended by the editor and the other two reviewers. The article is reading even better now. In my opinion, the paper deserves to be published. I recommend publication of this paper.

·

Basic reporting

The authors did an excellent job revising the manuscript and rewriting several paragraphs. As a result, the manuscript now looks more informative and thought-provoking. I believe it could be accepted for publication in PeerJ. As for the English grammar, the editorial office could decide whether the corrections made were sufficient or whether the text needs further revision.

Experimental design

ok

Validity of the findings

ok